


# Coastal circulation and eddies generation in the Southwest Mexican Pacific.

Federico Angel Velázquez-Muñoz[1], Raul Candelario Cruz-Gómez[1], and Cesar Monzon[1]

[1]Departamento de Física. Universidad de Guadalajara. Guadalajara, México.

**Correspondence:** F. A. Velazquez-Muñoz (federico.velazquez@academicos.udg.mx)

**Abstract.** In this work, we investigate if it is possible to identify the Mexican coastal current using satellite data and how this coastal current interacts with the coastline. This study is carried out using the mean sea level anomaly and derived geostrophic velocities from the Copernicus Marine Service from 1993 to 2021. By computing variance ellipses and empirical orthogonal functions for the ocean velocity data, we were able to identify and delineate the coastal current and determine its main characteristics: an average width of 95 km and an average velocity of 0.3 m s$^{-1}$. The preferred direction of this coastal current is parallel to the coast, with episodes towards the pole and towards the equators showing seasonal variability. Using a 3D numerical model, we found that the interaction of the coastal current with the coastline generates eddies near the coast in some places forming a wide concavity. The eddies are formed while the current is present (poleward or equatorward), become intense, and move away from the coast until the current weakens. The statistics of some physical variables of near-shore eddies show that they have a similar radius of 30 km and a vorticity of $0.5 \times 10^{-5} s^{-1}$. We conclude that there is a large number of eddies in the coastal zone of the Mexican Tropical Pacific, and some of these eddies are formed by the interaction of the coastal current with the coast.

## 1  introduction

Coastal currents are an important process in the oceans as they can transport physical properties such as heat, salinity, and biological properties such as larvae and chlorophyll, among others. These properties contribute to the sustainability of fishing regions and local ecosystems, and an adequate understanding of the dynamics of coastal currents goes a long way toward knowing how water masses are transported and characterizing the contribution at specific locations. On the eastern Pacific coast there are well-studied currents such as the Alaska Current (Stabeno et al., 2004) and the California Current in the north (Gan and Allen, 2005; Huyer, 1983), while towards the equator there are the Peru Current System (Brink et al., 1983), the Costa Rica Coastal Current and the Mexican Coastal Current (Badan-Dangon, 1998) that require further attention.

During the months of June and July, a coastal current that drives the coastal circulation develops in the Southwest Mexican Pacific (SMP), which has been called "The Mexican Current" or "West Mexican Current" (Wyrtki, 1965; Kessler, 2006). This coastal current was described by (Lavín et al., 2006) through observations made in June 2003 and June 2005. They found that this coastal current has a width of 90-180 km, a depth of 250-400 m and an average speed of 0.15-0.30 m s$^{-1}$. Recently, a study using a three-dimensional numerical model to analyze the seasonal variability of the coastal circulation off southwestern





Mexico shows that this current has a seasonal variability, being poleward/equatorial during summer/spring (gom, 2015). These authors mention that they found a connection between the Mexican Coastal Current and the Tropical Pacific, which is a topic that needs to be defined. There has been a longstanding interest in understanding this coastal current in order to determine its relationship to the general circulation of the tropical Pacific and the effects that climate variations, such as the El Niño

phenomenon, can have, as shown by (Terrazas Silva et al., 2024) using a 3D model.

Since we consider that the Mexican Coastal Current (MCC) it part of the dynamics of the SMP, is of particular interest to identify the current in satellite data, for example, in one of the sea surface height and geostrophic currents open access database available on the Internet. This type of measurement is an important tool to understand the global picture of the oceans and the connection between the different current systems (Klemas, 2012) since it is possible to know if physical properties such as

current temperature and salinity change and what local effects they have.

This work presents the results of detailed observations and analysis of geostrophic velocity data measured in the SMP with the aim of identifying the MCC and studying its interaction with the coastline. The methodology used allowed us to clearly identify the coastal current and the effect it produces in its interaction with the coastline, generating eddies that move towards the interior of the ocean. In the present study, we describe for the first time the existence of at least three sources places of

mesoscale eddies (cyclonic/anticyclonic) generation in the SMP and we described the main eddies characteristics of ratio and vorticity, being statistically similar. The paper is organized as follows. The data and methodology are presented in Section 2. The identification and interaction with the coastline of the coastal current are given in sections 3 and 4. Finally, the conclusions are presented in section 5.

## 2    Data and Methods

### 45    2.1    Study area and data

The region of interest is located on the southwest coast of the Mexican Pacific (Fig. 1a) between 254-264$^{\circ}$ east and 14-20$^{\circ}$ north (Fig. 1b). We are particularly interested in the coastal zone, which has an approximate extension of 1,130 km parallel to the coast, and for our purposes we consider a width of 200 km offshore (see the rectangle in Fig. 1b). The Global Ocean Gridded Sea Surface Heights And Derived Variables Reprocessed Copernicus Climate Service (cme, 2023) data set product

was used for this study. These data were subjected to all standard corrections and objective space-time interpolation to produce daily Sea Surface Height (SSH) and Mean Sea Level Anomaly (MSLA) fields on a $0.25° \times 0.25°$ grid. Absolute geostrophic flow $(u_g, v_g)$ and geostrophic flow component anomalies $(ua_g, va_g)$ in the zonal and meridian directions are also included in this database and are also used in this study. The msla and $(u_g, v_g)$ values were based on the years from 1993 to 2021 between 254$^{\circ}$ - 264$^{\circ}$W and from 14$^{\circ}$ - 20$^{\circ}$N. These data were processed and analyzed to identify and understand the coastal current

structure and to identify and describe the mesoscale eddies generated by the interaction of the MCC with the coastline. The components of the geostrophic velocity anomalies were obtained directly from the CMEMS product file data for subsequent analysis. The temporal mean current from the sla and $(ua_g, va_g)$ data and the polar histogram of the current direction at a point within the rectangle where the coastal current develops are shown in Figure 1b. It is clear from the polar histogram that



the main direction of the currents is aligned with the coastline and has two main modes; poleward and equatorward. Another
important aspect to note in Figure 1b is that there are two anticyclonic circulation zones, the first off Manzanillo, Colima (MZ)
and the second off Punta Maldonado, Guerrero (PM), which we will discuss later.

## 2.2 Eddie identification

The identification of ocean eddies and their physical properties were obtained using the Angular Momentum Eddy Detection
and Tracking Algorithm (AMEDA) (Vu et al., 2018). This algorithm is based on a combination of physical and geometrical
properties of a potential vortex in a velocity field. The identification process starts with the location of the center of the vortex,
which corresponds to a maximum of the local normalized angular momentum and a closed streamline around it. This method
has been shown to be highly effective in detecting mesoscale eddies generated by coastal current instability (Ioannou et al.,
2017; Aroucha et al., 2020). The main AMEDA output parameters for each eddy are: the center coordinates ($x_o$, $y_o$ in degrees),
the maximum radius ($r_{max}$ in meters), and the relative vorticity ($\omega$ in cycles per second). AMEDA delimits $r_{max}$ according
to the module of the maximum azimuthal velocity ($v_{max}$ in meters per second), which is derived by taking values of each
interpolated streamline from the velocity field. With its dynamic properties, AMEDA is able to find and track the evolution of
vortices and their lifetime. This data processing with AMEDA software was used to find eddies after we identified the coastal
current.

## 3 Mexican Coastal Current in the SMP

In order to identify spatial patterns of coastal current direction in the anomaly of geostrophic velocity, we calculated the
parameters of variability ellipses (Signell, 2000) in the area of interest, and thus identify the degree of orientation of the
preferred current direction. The first step is to compute the anomaly of the velocity time series given by $u'(t) = u(t) - <u>$,
$v'(t) = v(t) - <v>$, where $(u(t), v(t))$ is the original time series and $<u>, <v>$ are the time averages. We use Rich
Signell's (2014) RPSstuff set of routines to compute the major axis ($r_{max}$) and the minor axis ($r_{min}$). Once we have the
amplitude values of the principal axes, we calculate the eccentricity by $\epsilon = 1 - r_{min}/r_{max}$. We are looking for the nodes
where the eccentricity is greater than 0.3, which indicates less dispersion in the current direction, and in our case, when the
eccentricity is close to 0.0, it indicates a large dispersion in the current direction. We are interested in finding the nodes where
the velocity has a high eccentricity and the orientation of the main axis is parallel to the coast, since this corresponds to the
nodes where the coastal current develops. In figure 2a, the grid nodes where the time series current produces ellipses with
$\epsilon$ equal or greater than 0.3 are shown in blue. It can be seen that only the nodes near the coast show this behavior, while the
offshore nodes with $\epsilon < 0.3$ have a high dispersion in the current direction. Apparently, the eccentricity of the current variability
is a good method to identify the nodes of interest in front of the coast, and for this reason we take these nodes (with $\epsilon > 0.3$)
that are below $20°$N and to the left of $263°$E. These nodes are in front of the coastline where coastal currents develop (see
black arrows in Fig. 2a).



The next step to study the temporal changes and spatial patterns of the current variability was to compute the Empirical Orthogonal Functions (EOFs) for the time series of the current velocity in front of the coast. To compute the EOFs, we use a computationally efficient method based on singular value decomposition (SVD). Using the same velocity anomaly time series $(u'(x,y,t), v'(x,y,t))$, for the nodes where the eccentricity of the variability ellipse is greater than 0.3, we construct a matrix array $W = W_{ij}$, where each j-th column is a time series $w(t) = u'(t), iv'(t)$. We obtain a first mode that explains 33.7% of

the variability and captures the flow that develops parallel to the coast. This value for the first mode is larger than the 22% reported by gom (2015) for results from a numerical model, but they consider all numerical nodes of the model domain they used. The vector field associated with the first mode shows a well-defined current parallel to the coast with an approximate width of 84 to 107 km. The temporal variation of this first mode is consistent with that reported by gom (2015) and others, with a seasonal variation that produces a current toward the pole in summer and toward the equator in other seasons. For this

first mode of EOFs, when the current is toward the equator, the average velocity is 0.13 ms$^{-1}$ with a maximum value of 0.27 ms$^{-1}$, while when the velocity is negative (poleward), the average is 0.16 ms$^{-1}$ and the maximum value is 0.33 ms$^{-1}$. So far, we have managed to identify the coastal current in satellite data, which is one of the most important tools for understanding synoptic ocean conditions over large areas (Klemas, 2012).

## 4    Generation of coastal eddies

During the processing and visualization of the geostrophic velocity, we observe the emergence of many eddies, which could be called current instabilities and were described by Zamudio et al. (2001). However, a more detailed observation of the velocity fields shows that there are at least three locations in the SMC where more small eddies emerge. This process of eddy formation is different from what has been described by Zamudio et al. (2001) because these eddies form between the coastal current and the coastline. Figure 3 shows four maps for the days of July and August of 2008. In these maps, the colors at MSLA and the

black arrow correspond to the geostrophic velocities. In the July 5 map (Fig. 3a), a well-developed poleward coastal current is observed along the coast of the SMP. On July 14, the near-coastal current begins to develop as an anticyclonic circulation at three locations indicated as Manzanillo (MZ), Lazaro Cardenas (LC), and Punta Maldonado (PM) in figure 3b. Ten days later, on July 24 (fig. 3c), there is a well-developed clockwise eddy off MZ and a less developed clockwise eddy off PM, close to the coast. On August 14 (Fig. 3d), the eddy that formed in front of MZ has moved offshore and is surrounded by the remnants of

the coastal current, while the eddy in front of PM is now well developed. On the same maps, there is a larger eddy at 257°W and 15.5°N, which appears to originate from LC, but we cannot be sure. We check the sequence of several dates where the coastal current is developed poleward and equatorward off the coast of the SMP, and we find that it is also common to observe the formation of these eddies between the coastline and the coastal current.

Considering the whole time period from 1993 to 2021 of the data, we use the AMEDA software to identify and track all the

eddies that appear near the coast. From the AMEDA output, we consider the center position and the direction of rotation as the first classification feature. In figure 4 we plot the eddy centers and trajectories according to their sense of rotation in red for counterclockwise and blue for clockwise. The plots in figure 4 are separated by season. The blue and red dots correspond to the





first center, which identifies the eddies with more than seven days of life. These maps show that eddies are generated near the coast in all seasons and can be cyclonic or anticyclonic. In both cases, the path of the eddy is southwest, away from the coast.

In these figures 4 it is possible to see that the starting points of the eddies are distributed along the entire length of the coastline considered. Nevertheless, a slight accumulation of eddy origins is observed in front of three locations where the coastline has a concavity, so we can assume that some of these eddies are formed by the interaction of the current with the coastline. It is not easy to show the above because there is a large number of eddies in this area and it requires a more extensive processing of the data, which is not the objective of this work.

Using all the AMEDA output data, we separate the monthly eddies by their sense of rotation for all available years. Figure 5 shows the number of cyclonic and anticyclonic eddies found for each month over the entire data period, red for cyclonic and blue for anticyclonic. In figure 5b we show the time series of the coastal velocity component for each year (gray lines) and the black line is the daily mean of the coastal velocity component. As we can see, in March and April there are more cyclonic eddies associated with the equatorward coastal flow, and in June and July there are more anticyclonic eddies associated with the poleward coastal flow (Fig. 5a). For the remaining months (January, February, May, August through December), a comparable

number of cyclonic and anticyclonic eddies are present. Two of the main characteristics of the observed near-coastal eddies are the radio and the vorticity. These two variables allow us to characterize these eddies, and allow us to identify if there are differences between the eddies generated in each of the three places. Figures 6 show two histograms for radii (Fig. 6a) and vorticity (Fig. 6b). As in the previous figure, the red color is for cyclonic eddies and blue for anticyclonic eddies. Most of the

near-shore eddies have a radius of 30 km, with some eddies having a radius of 20 or 40 km (Fig. 6a). This radius is the first time the vortices are well defined. In the same sense, the vorticity is well defined, with values of $1.0$ or $0.5 \times 10^{-5}$ s$^{-1}$ (positive for cyclones and negative for anticyclones) for most of the eddies (Fig. 6b). These statistical results show that there is a great consistency in the physical properties of the identified eddies. Therefore, we can define that the coastal current interacts with the coastline to generate submesoscale eddies with a typical radius of 30 km and a typical relative vorticity of $0.5 \times 10^{-5}$ s$^{-1}$.

**5 Numerical model**

In order to explain the formation of eddies due to the interaction between the current and the coastline, we implemented the numerical model POM (Princeton Ocean Model) for the study area (Blumberg and Mellor, 1987). The POM is a 3D numerical model that solves the nonlinear primitive equations under the Boussinesq approximation and the hydrostatic approximation using finite differences. The domain of the model corresponds to the area of the rectangle in figure 1b. The mesh has 701 x 126

nodes and a horizontal resolution of 2.2 km, so that the coast of Mexico can be covered (see figure 7a). Vertically, the model uses 41 layers in sigma coordinates. The bathymetry for the model was taken from the ETOPO Global Relief Model database (Amante and Eakins, 2009) with a resolution of 1 arc minute in longitude and latitude and interpolated to the rectangular grid using the Arakawa type C configuration. The initial velocity condition is quiescent, with velocity and sea surface level equal to zero. The only forcing imposed on the model is a velocity parallel to the coast at the lateral extremes, i.e., to the right and

left of the domain. In this way we can reproduce in the model the coastal current towards the pole (when it goes to the left) and




the coastal current towards the equator (when it goes to the right). The temporal variation of the current shown in the figure (7b) consists of an increase from 0.0 to 0.25 m/s in a period of 7 days, then remaining constant at 0.25 m/s for 14 days and decreasing in another 7 days until reaching a value of 0.01 m/s to maintain this value until the end of the 30 day simulation. This velocity variation can be applied positively or negatively to simulate the flow to the pole (right) and the equator (left).

The model output data that we look at to observe the interaction between the current and the coastline are the components of the surface velocity ($u_s$, $v_s$). The model output data are shown in velocity maps for the whole numerical domain and for three locations identified as the main ones for eddy generation.

The figure (8) shows the case when the current is poleward for day 13 (top panel) and day 28 (bottom panel) of the simulation. Above each panel, three frames are shown with a close-up of the areas identified as eddy generation. In the figure (8 top panel)

at (b) and (c) it is possible to observe the onset of anticyclonic eddy generation. A map of the surface flow over the entire domain is shown in the bottom of each panel.

Similarly, figure 9 shows the case where the current is directed towards the equator. In the figure (9 top panel), on day 11 of the simulation, the well-developed coastal current is observed and the formation of a cyclonic circulation begins in the LC and PM regions. In the MZ region, no cyclonic circulation formation is observed during the formation of this current towards the

equator. On day 28 of the simulation (9 bottom panel), the two cyclonic structures in front of LC and PM persist, although the speed of the current has already decreased considerably.

These results in the coastal circulation from the numerical model show that the coastal current separates from the coast at the three locations shown in Figures 8 and 9, called MZ, LC and PM. In these areas, the current separates from the coast, creating recirculation zones that become eddies.

**6 Conclusions**

For this work, we use the data from the SSALTO/Duacs Altimetric from Copernicus Marine Service (CMEMS) product with the main objective of identifying the coastal current in the southwestern Mexican Pacific and studying its interaction with the coastline. The data product provides the sea level anomaly and the components of the geostrophic velocity anomalies. The data period used is 29 years, from 1993 to 2021. By calculating the eccentricity of the variability ellipses and the empirical

orthogonal functions both for the velocity field, we found that it is possible to identify the coastal current and that it has the same characteristics that have been reported by observations (Lavín et al., 2006) and numerical models. (gom, 2015).

The variability ellipses of the velocity currents allow us to find nodes where the current has a preferential direction parallel to the coast, in such a way that they are representative of the coastal current. Then, by calculating the EOFs of the currents in the nodes where the eccentricity of the current variability ellipse is greater than 0.3, we obtain in the first mode the temporal

and spatial variability of the coastal current. Detailed observations of the evolution of the coastal current helped us to identify that there is an interaction between the coastal current and the coastline. When the coastal current develops, eddies are formed between the coastal current and the coastline, at least in three specific places. These specific places have variations of the coastline that form sea inlets into the land, forming a wide concavity that allows eddies to form. When the current is toward





the pole, the eddies that form are anticyclonic, and when the current is toward the equator, the eddies that form are cyclonic. In general, the eddies that form near the coast have similar radii and vorticity, but it is difficult to separate or identify the origin of so many ocean eddies. We propose that there are variations along the coastline where conditions are favorable for the formation of these eddies.

Under this assumption, the formation process of these eddies can be explained in three steps: 1) the coastal current develops and begins to generate recirculation where there are concavities; 2) the coastal current induces anticyclonic/cyclonic circulation in some places when its direction is toward the pole/equator; and 3) when the current weakens and disappears, the eddies move away from the coast toward the open sea. This process of eddy formation must have inter-annual variability, since the impulse that the eddies receive depends on the intensity and life of the current. Once we identify the eddies, the analysis of the position of the eddy center shows that the three locations along the southern coast of Mexico where most of the eddies originate are Manzanillo (MZ), Lazaro Cardenas (LC), and Punta Maldonado (PM). These coastal eddies have a short life span of a few weeks as they move towards the open ocean. Most of the eddies forming near the SMP coast have a radius of 30 km and reach a maximum relative vorticity of $1.0 \times 10^{-5}$ s$^{-1}$. Finally, we want to emphasize that the generation process and the temporal evolution of the observed eddies need to be addressed by theoretical and numerical models and by direct observations. In order to prove whether eddy generation is associated with concavities in the coastline, we used a simple simulation with a numerical model.

The generation process of these eddies by the interaction between the coastal current and the coastline was studied using the 3D POM model, imposing a current towards the pole and towards the equator. The results show that when the current develops there are three places where the current separates from the coast, giving rise to a recirculation that later became as ocean eddies. A difference we found with the model is that the poleward current can generate eddies in all three locations, while the equatorward current can only do so in two locations, LC and PM. In summary, the model results show that the interaction of the coastal current with the coastline produces eddies at specific sites, while the observations show a large number of eddies and we cannot be sure of the origin of all of them. There are many other processes that produce eddies, such as the instability of the current when it weakens (Zamudio et al., 2001), but only the eddies that form between the current and the coastline are due to interaction. Since the statistical results of the characteristics of the eddies in the region are similar, it is difficult to identify a characteristic that allows the identification of the eddies generated by the interaction of the current with the coast. In addition, the formation and evolution process is at the sampling limit of the data used, so other types of studies, such as local sampling, are required to verify the dynamics of these eddies.

These eddies play an important role in biological and chemical processes that significantly affect local ecosystems, as has been studied in the California Current, for example (Kurian et al., 2011). As for the Mexican Coastal Current, to our knowledge, there are no detailed studies on the local effects of the coastal currents, and the studies that do exist focus on studying the variability of the current (gom, 2015) and the processes of generation and connection with the tropical Pacific (Terrazas Silva et al. (2024)).



*Data availability.* The Mean Sea Level Anomaly (msla) and the Absolute Geostrophic Current ($u_g$, $v_g$) were obtained from the DUACS delayed-time altimeter gridded maps of sea surface heights and derived variables over the global (https://doi.org/10.48670/moi-00145, 2023) produced and distributed by the Copernicus Climate Change Service (C3S, https://climate.copernicus.eu/, last accessed: March 225  2023). Bathymetry data were obtained from the ETOPO1 1 Arc-Minute Global Relief Model (Amante, C. and B.W. Eakins, 2009.) Procedures, Data Sources and Analysis. NOAA Technical Memorandum NESDIS NGDC-24. National Geophysical Data Center, NOAA. doi:10.7289/V5C8276M [accessed August 18, 2023].

*Author contributions.* FAVM and RCCG developed the methodology for data processing and FAVM, RCCG, and CM contributed to the interpretation of the results, discussion, and with the writing of the draft manuscript.

*Competing interests.* The contact author has declared that none of the authors has any competing interests

*Acknowledgements.* We thank Reginaldo Durazo for his valuable comments on the manuscript.



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



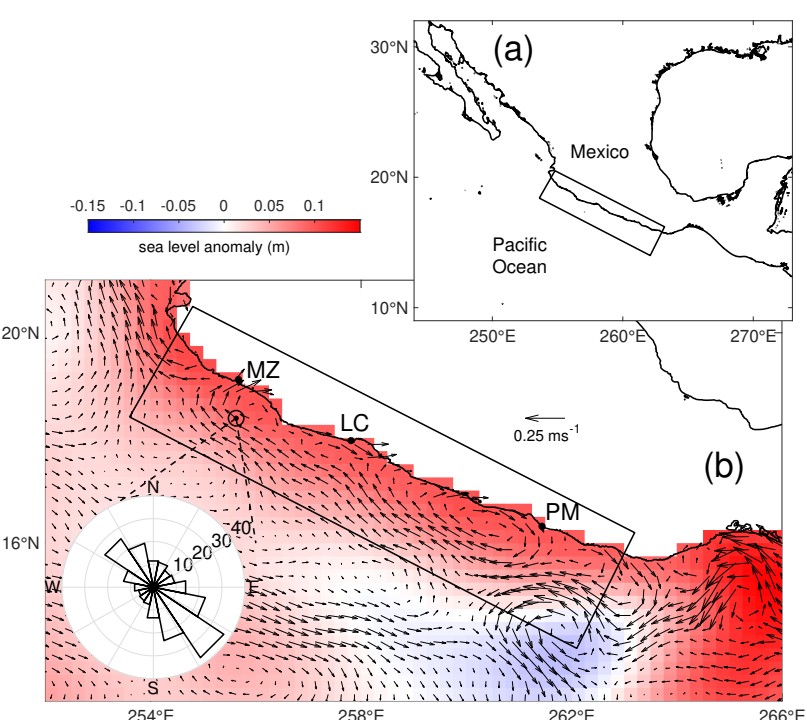

**Figure 1.** Map of the study zone (black rectangle) in the southwest coast of Mexico (a) and mean velocity field from 1993 to 2021 with the polar histogram of velocity at one grid node at the core of the WMC (b).





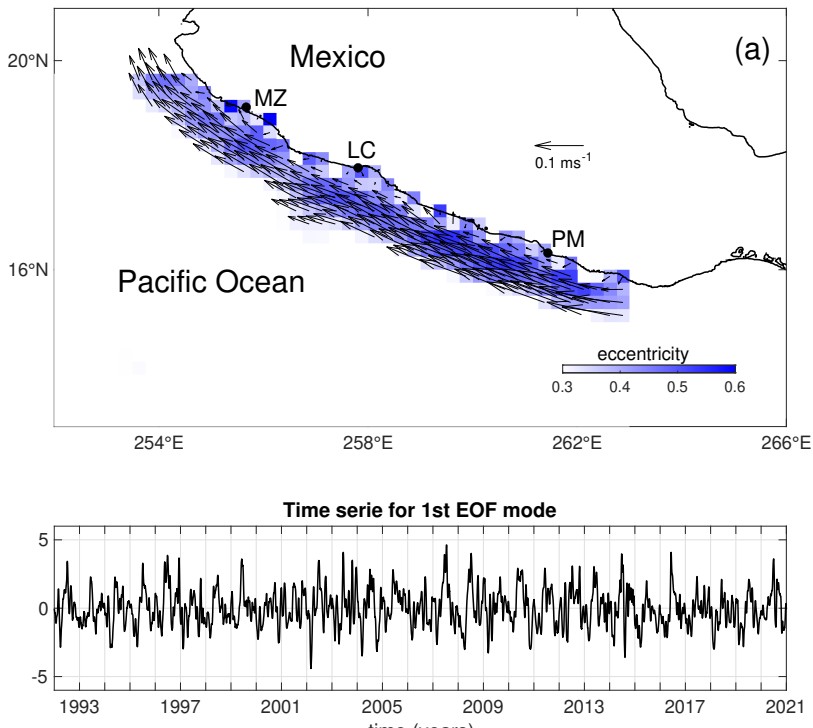

**Figure 2.** Representation of the firt EOF mode as (a) map of the vector field (espacial mode) and (b) time series (temporal mode). The blue tones nodes correspond to velocity time series with $\epsilon > 0.3$



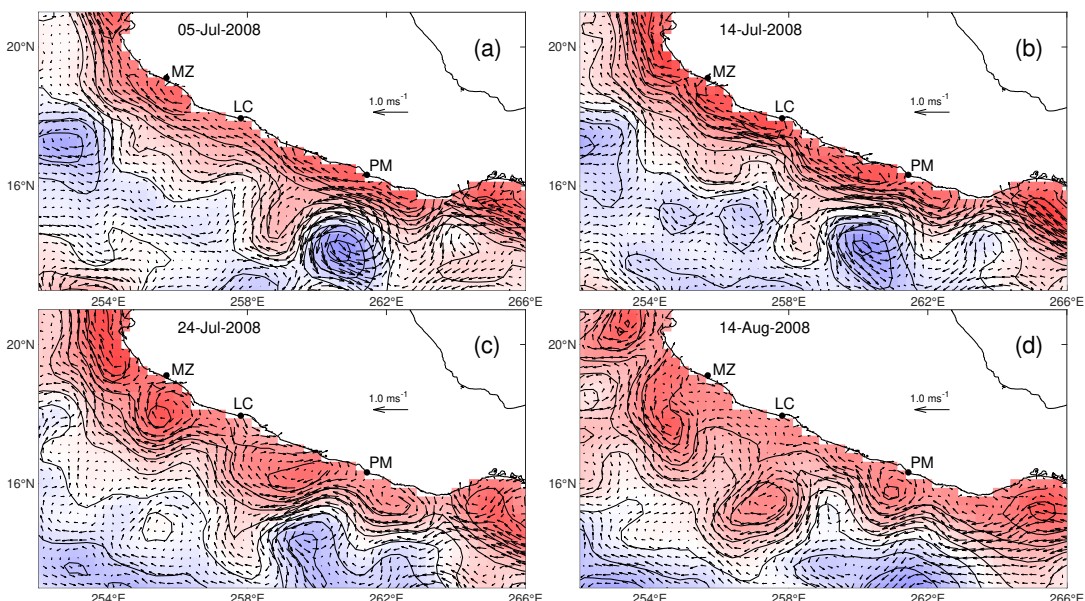

**Figure 3.** Geostrophic velocity (in m s$^{-1}$) derived from SSHA (blue/red tones) for some days in July and August of 2008 showing the current flow close to the coast and the evolution of the recirculation as the current begins to weaken taking off from the coast and allowing the eddies to move freely.





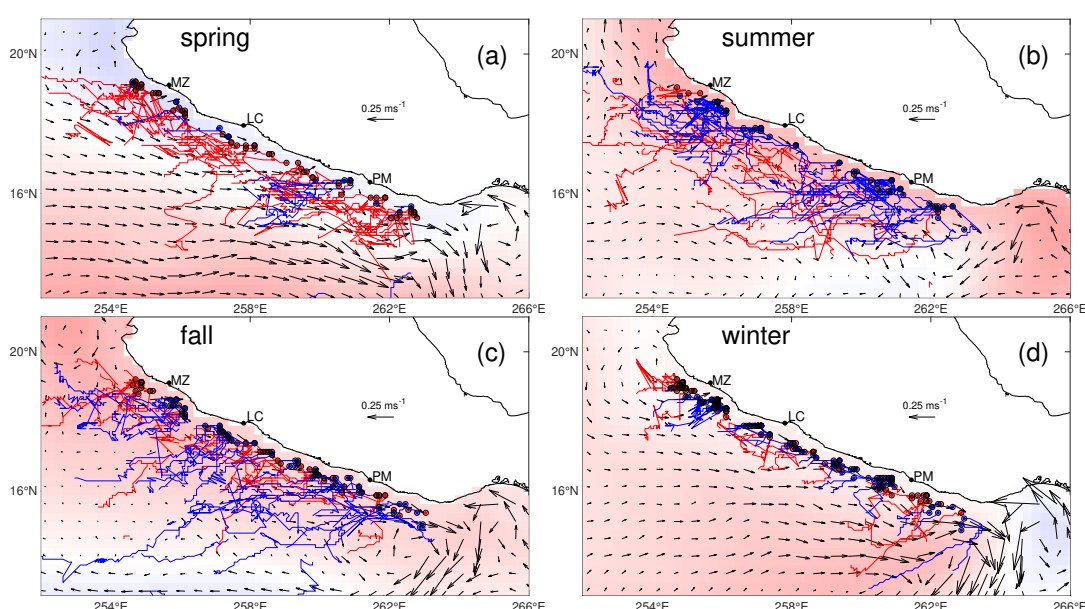

**Figure 4.** Initial points and paths of identify coastal eddies generated in the Southwest Mexican Coast. The color and vectors correspond to SSHA and geostrophic velocity, respectively derived from **CMEMS** data.



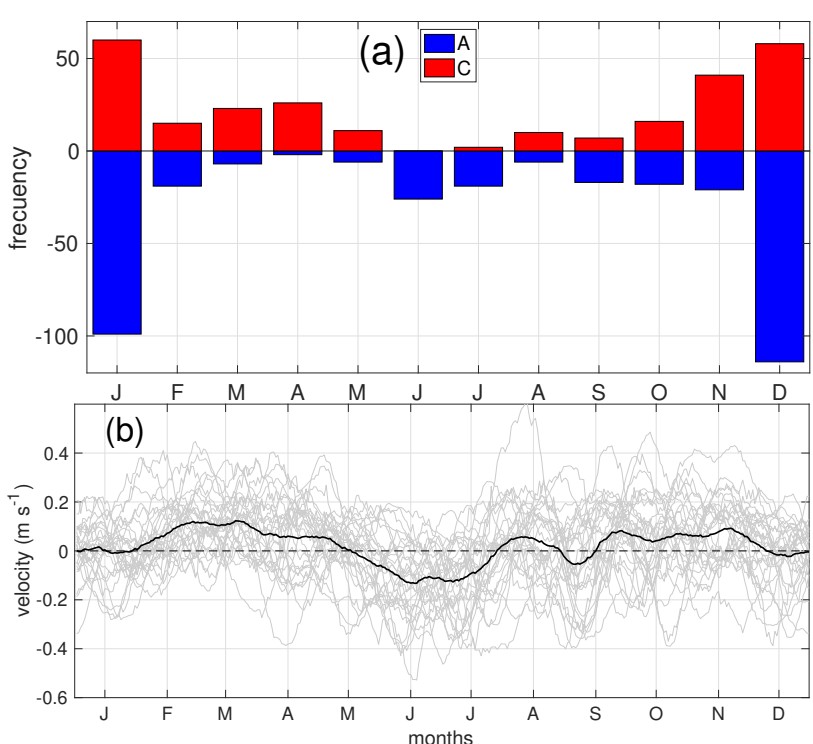

**Figure 5.** (a) Monthly climatology from 1993 to 2021 of emerged cyclon/anticyclon eddies in red/blue. (b) Annual time series of component velocity along the coast for each year (gray) and daily climatology from 1993 to 2021.



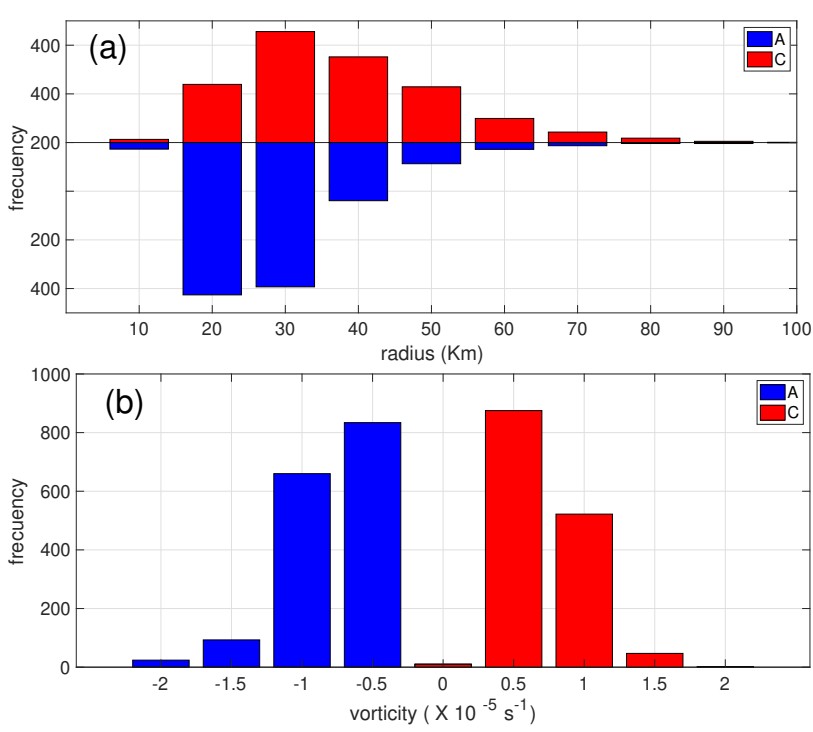

**Figure 6.** Histogram of vortex radio (a) and relative vorticity (b). In both, red/blue are for cyclonic/anticyclonic ocean eddies.



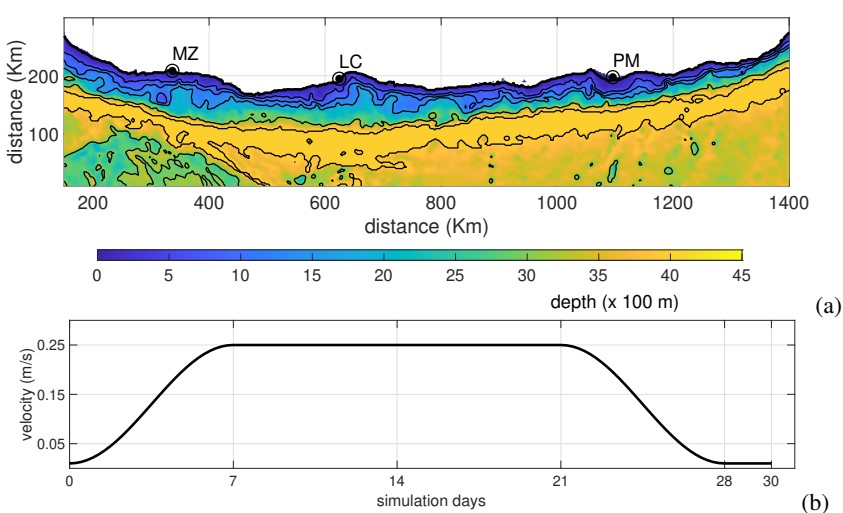

**Figure 7.** (a) Numerical model domain and (b) coastal current velocity modulation



**Figure 8.** Poleward coastal current (to the left) for day 13 (top panel) and day 28 (bottom panel) of numerical simulation. In both panels show details of the concavity places where the eddies are formed and of the entire domain.



**Figure 9.** Equatoreward coastal current (to the right) for day 11 (top panel) and day 28 (bottom panel) of numerical simulation. In both panels show details of the concavity places where the eddies are formed and of the entire domain.