# Peer review of "Coastal circulation and eddies generation in the Southwest Mexican Pacific."

_EGUsphere, 2024_

## Author Comment (AC1)

**RC1**: 'Comment on egusphere-2024-3403', Anonymous Referee #1, 15 Jan 2025

This is the review of the paper entitled "*Coastal circulation and eddies generation in the Southwest Mexican Pacific*" by F. Velázquez-Muñoz, R. Cruz-Gómez, and Cesar Monzon. The paper uses satellite data and numerical modelling to study the variability of the Mexican coastal current and eddy generation in the Pacific Tropical Mexican.
Although the characterization of Mexican coastal current is somewhat known with previous works, I believe that the novelty of this work is the generation of eddies in the bays driven by the interaction of the current and the geometry of the coast.

I do not see what the new findings regarding the characteristics and variability is of the coastal current in this analysis that has not been shown before? The bimodal structure and variability are already known.
**Yes, we agree. The variability of the current is known, but this variability of the coastal current that is known has been observed with output data from numerical models and in our work, it is with data from remote measurements. We wrote about this in the manuscript to make it clearer. Thank you by the comment.**

However, I believe that this work could be still published if some additional work is performed.
Specifically, the generation mechanism of the eddies is poorly supported and just described superficially.
**It is not an objective of the work to make a robust model, but rather we are only trying to explain whether it is possible for a coastal current to form eddies when interacting with the coastline.**
**For a better description of the circulation of this region of the Pacific, a simulation more appropriate to the conditions of hydrography and environmental currents is needed and that is not our objective for now. For now, we need a very controlled simulation to understand whether the coast-current interaction produces the eddies.**
**Considering your comment, we include in the manuscript the relative vorticity, calculated with the surface velocity. We believe that this variable helps considerably to answer this comment about the mechanism of eddy formation.**

To provide insights on mechanism of generation, the authors could use the numerical outputs to compute the terms for the material rate of change of the vertical vorticity and inspect what are the dominant forcings that "inject" vorticity

in the bays were the eddies are generated. I encourage the authors in do so this analysis to conclude about the generation mechanisms of the observed eddies.
**We now include the data from the relative vorticity calculation to show that the coastal capes or point produce an acceleration of the current, increasing the relative vorticity and the current carries it towards the concavity zones where eddies are formed by the accumulation of vorticity. We hope this point is clearer.**

The paper needs additional computations, and the English grammar also needs to be revisited.
**We agree that grammar needs to be revised, and we are going to take advantage of the editor's offer. We are already in communication with the responsible department as shown in the following emails.**

*Copernicus Publications*
*The Innovative Open-Access Publisher*

*Natascha Töpfer*
*Editorial Support | Team Coordinator*

*This is an email received in 13 nov 2024, 6:40:*
*Our copy-editors have checked the English of your manuscript and decided that a copy-editing is not needed at this stage since the English is well enough for your manuscript to be posted as a preprint. Of course, if finally accepted your paper will receive a full copy-editing as part of our standard service. The editor is already informed, and you can upload the revised version as soon as you have taken all other comments from the editor into account.*
*Copernicus Gesellschaft mbH*
*USt-IdNr.: DE216566440*
*Based in Göttingen, Germany*
*Registered in HRB 131 298*
*County Court Göttingen*
*Managing Director Thies Martin Rasmussen*

For all that I recommend a major revision. Please find detailed comments below.
**Line 26:** gom (2015) is not referenced in bibliography. Please include the reference.
**Thank you very much for the comment. We reviewed all the references and made the necessary changes.**

**Line 31:** replace "it" by "is"
**Attended**

**Line 25:** current, temperature, and salinity (use commas)
**Attended**

**Lines 38-39:** "towards the interior of the ocean" --> westwards
**Attended as:**
**Replace:**
*The methodology used allowed us to clearly identify it and the effect it produces in its interaction with the coastline, generating eddies that travel towards the interior of the ocean.*
**By:**
*The methodology used allowed us to clearly identify it and the effect it produces in its interaction with the coastline, generating eddies that travel offshore.*

**Line 53:** msla should be uppercase as was defined the acronym.
**Attended.**

**Line 57:** "the current direction at a point". Why choose a single location for polar histogram of current? The authors should average over the entire region of interest or over a larger area since the current at a single location can be very variable, for example, could influenced by the presence of an eddy or meandering jets.
**That is correct. It is not the best option to explain this matter of the preferential direction of such an extensive process with a single point. In our experience in the study of this region, and in particular for this current, we calculate several histograms or direction roses and obtain similar results. We want to show only the histogram at one node as an example. In any case, in the manuscript we explain that it is representative of the others, and we attach next figures as evidence.**

[Figure]

**Line 61:** The second eddy is located between LC and Punta Maldonado (PM), Guerrero.
**We replace by: The second eddy is located between Lazaro Cardenas (LC), Michoacan and Punta Maldonado (PM), Guerrero.**

**Line 65:** Please define "potential vortex"??
**It refers to a potential vortex, that is, a possible vortex in the velocity field of the current.**
**We replace:**
**This algorithm is based on a combination of physical and geometrical properties of a potential vortex in a velocity field.**
**By this:**
**This algorithm is based on a combination of physical and geometrical properties of a potential vortex, that is, a circular flow around a central point.**

**Line 78:** replace "is" by "are"
**Attended.**

**Line 79:** "rmax". Is this rmax a radius as defined by AMEDA or the semi-major axis of the ellipse? Please clarify.
**Attended. We used R_max for AMEDA ratio and r_max for semi-major axis**

**Line 80 and throughout the paper:** what the authors mean by "nodes"? Are "locations"?
**We now clarify in the manuscript that nodes are locations in the data grid.**

**Line 81:** "greater than 0.3". What is the reasoning to choose 0.3m/s? This selection would widen or shorten the region driven by along-coast "unidirectional" current . Please clarify.
**Once we had calculated all the eccentricity values at each node in the data grid, we made a map of the eccentricity value and noticed that there are high values in front of the coast compared to the other nodes far from the coast. Then, we discriminated the lower values (greater variability) until the value of 0.3 or higher allowed us to isolate the nodes in front of the coast, as shown in figure 2 of the manuscript.**
**It should be noted that the current direction roses were reviewed at several nodes in the data grid and we found that they are bidirectional, oriented parallel to the coast. The clarification is made on lines 163 and 164 of the manuscript.**
**Some illustrative figures are included here.**

**Line 83:** "main axis" is the semi-major axis.
**Attended. We replace "main axis" by "semi-major axis"**

**Line 87:** "in front of the coast" refers to offshore?
**No, it refers to the nodes of the data grid that are located near the coast or in front of the coast, which are the nodes that we identify as nodes where the current develops.**

**Line 94:** Is this a time series of complex number? Define i=sqrt(-1) and rewrite the equation w(t)=u(t)' + i*v(t)'.
**Attended as it.**

**Line 107:** "shows" --> "show" and "more small" --> "smaller"
**Attended**

**Line 113:** "clockwise eddy" --> "Anticyclone"
**Attended**

**Line 121-122:** This tabulation for colors is bit confusing. Typically, red is used for (warm) anticyclones and blue for (cold) cyclones…but ok
**We took the sea level variable as a reference for these colors, but we agree that it can be changed if necessary for a better understanding of the work.**

**Line 126:** "locations where the coastline has a concavity" --> bays.
**Well, as far as we have investigated, these sites are not bays.**

**Line 135:** "For the remaining months (January, February, …)". I do not see comparable number of cyclonic and anticyclonic eddies. Note that in January and December there are 100/50 anticyclonic/cyclonic eddies (2 times more). Or in November 20/40 anticyclonic/cyclonic (2 times less) but not comparable. What I see is a clear relationship between poleward current and anticyclone generation, equatorward current and cyclone generation.
**Yes, the relationship between eddies is not comparable and there are months in which there are eddies in both directions. It is correct that when the coastal current develops, the respective eddies are formed.**

**Line 144:** "submesoscale eddies" -->  You are not resolving the submesoscale eddies as observed in the PDF of the radius (Fig. 6a). There is a strong drop of the frequency below radius of 20km. This is because the resolution of the data is 25 x 25km. You need to use SWOT data (higher resolution ssh and geostrophic velocity data; however, the temporal resolution is of 15-20 days so not adequate for this kind of study). I would call mesoscale eddies.
**I agree. We changed to "submesoscale" since it is correct.**

**Paragraph 156-161:** Is this temporal behavior based on actual observations. The authors should compare with the temporal modulation of the satellite-derived geostrophic current.
**On the data website it says the following:**
**Overview**

DUACS delayed-time altimeter gridded maps of sea surface heights and derived variables over the global Ocean (https://cds.climate.copernicus.eu/cdsapp#!/dataset/satellite-sea-level-

). The processing focuses on the stability and homogeneity of the sea level record (based on a stable two-satellite constellation) and the product is dedicated to the monitoring of the sea level long-term evolution for climate applications and the analysis of Ocean/Climate indicators. These products are produced and distributed by the Copernicus Climate Change Service (C3S, https://climate.copernicus.eu/).

**DOI (product):** https://doi.org/10.48670/moi-00145

**Paragraph 163-171:** The days selected in figures 8 and 9 are arbitrary. Why choose times where zero-acceleration? What occurs during acceleration or deceleration of the coastal current? Please discuss the differences between acceleration/deceleration and steady current (zero acceleration).

**The selected dates show the eddies already formed.**
**A better explanation/discussion of the eddy formation process (acceleration) and dissipation process (deceleration) was included in the manuscript.**

Also, I am wondering whether this numerical analysis is showing the generation mechanism (?): (1) is the result of potential vorticity conservation for an equivalent barotropic flow?  (2) is the result of propagation of low-frequency coastal waves, (3) current instabilities…The analysis of the terms of the vorticity equation would provide insights on generation mechanisms: vortex stretching, tilting, advection of vorticity, …

**According to the results we obtained from the numerical model, the formation of eddies can be explained as the interaction of the coastal current with the coastline, where the capes produce acceleration of the current with an increase in vorticity that is dragged towards the concavity zones, where the current moves away from the coast, a cyclonic or anticyclonic circulation is formed depending on the direction of the current and there is an accumulation of vorticity. We wrote an explanation in the numerical model section and we hope that now we make this idea clearer.**